# Associations between Measured and Patient-Reported Physical Function and Survival in Advanced NSCLC

**DOI:** 10.3390/healthcare10050922

**Published:** 2022-05-17

**Authors:** Kristin Stokke, Tarje Onsøien Halvorsen, Bjørn Henning Grønberg, Ingvild Saltvedt, Marit Slaaen, Øyvind Kirkevold, Kristin Toftaker Killingberg, Marie Søfteland Sandvei

**Affiliations:** 1Department of Clinical and Molecular Medicine, NTNU—Norwegian University of Science and Technology, N-7491 Trondheim, Norway; tarje.halvorsen@gmail.com (T.O.H.); bjorn.h.gronberg@gmail.com (B.H.G.); kristin.t.killingberg@ntnu.no (K.T.K.); marie.s.sandvei@ntnu.no (M.S.S.); 2Department of Oncology, St. Olavs Hospital, Trondheim University Hospital, N-7006 Trondheim, Norway; 3Department of Geriatrics, St. Olavs Hospital, Trondheim University Hospital, N-7006 Trondheim, Norway; ingvild.saltvedt@ntnu.no; 4Department of Neuro Medicine and Movement Science, NTNU—Norwegian University of Science and Technology, N-7491 Trondheim, Norway; 5The Research Centre for Age Related Functional Decline and Diseases, Innlandet Hospital Trust, P.O. Box 68, N-2313 Ottestad, Norway; marit.slaaen@sykehuset-innlandet.no (M.S.); oyvind.kirkevold@aldringoghelse.no (Ø.K.); 6Institute of Clinical Medicine, University of Oslo, N-0318 Oslo, Norway; 7The Norwegian National Centre for Ageing and Health, Vestfold Hospital Trust, N-3103 Tønsberg, Norway; 8Department of Health Science in Gjøvik, NTNU—Norwegian University of Science and Technology, N-2802 Gjøvik, Norway

**Keywords:** physical performance, timed up and go, 5-meter walk test, advanced NSCLC, chemotherapy, overall survival

## Abstract

Background: There is a lack of tools for selecting patients with advanced lung cancer who benefit the most from systemic treatment. Patient-reported physical function (PRPF) has been identified as a prognostic factor in this setting, but little is known about the prognostic value in advanced non-small-cell lung cancer (NSCLC). The aim of this study was to investigate if measured physical performance was an independent or stronger prognostic factor than PRPF in patients with advanced NSCLC receiving platinum-doublet chemotherapy. Methods: We analyzed patients from a randomized trial comparing immediate and delayed pemetrexed therapy in stage III/IV NSCLC (n = 232) who performed timed up and go (TUG) and 5 m walk test (5 mWT) and reported physical function on the EORTC QLQ-C30 before chemotherapy commenced. Results: Overall, 208 patients performed TUG and 5 mWT and were included in the present study. Poor physical function was significantly associated with poor survival (TUG: HR 1.05, *p* < 0.01, 5 mWT: HR 1.05, *p* = 0.03, PRPF: 1.01, *p* < 0.01), but only PRPF remained an independent prognostic factor in multivariable analyses adjusting for baseline characteristics (HR 1.01, *p* = 0.03). Conclusions: Patient-reported, but not measured, physical performance was an independent prognostic factor for survival in patients with advanced NSCLC receiving platinum-doublet chemotherapy.

## 1. Introduction

Lung cancer is the third most common cancer and the most common cause of cancer-related deaths [1]. About 40% of patients have advanced disease at the time of diagnosis with limited survival expectancy and are offered palliative, systemic treatment [1]. Immune checkpoint inhibitors (ICI) and targeted therapies have improved survival for patients with advanced non-small-cell lung cancer (NSCLC), but cytotoxic chemotherapy still has a role, alone or combined with ICIs [2]. Even if chemotherapy is usually reserved for patients with a good performance status [3], response rates are moderate, approximately 30–35%, and it would be of great value to identify the patients who benefit the most from such therapy.

There is evidence that patients with poor physical function experience more toxicity from treatment and are consequently less able to complete treatment as planned [4], and several studies have shown that patient-reported physical function (PRPF) is an independent prognostic factor in advanced NSCLC [5,6]. Furthermore, there are indications that lower extremity function reflects patients’ health status and is prognostic in patients with cancer [7,8]. Timed up and go (TUG) [9,10] and gait speed are simple yet sensitive measures that have consistently been identified as prognostic factors among patients with cancer [7,8]. However, previous studies have included patients with different cancers, stages of disease, and treatment, and only one study adjusted for other important prognostic factors such as performance status (PS) in the analyses [11]. Consequently, there is limited knowledge of their independent prognostic information, and it is unclear whether these measures provide more clinically relevant prognostic information than PRPF. Additionally, if patients with poor physical function tolerate less systemic therapy, they might achieve less disease control. However, no study has investigated whether there are associations between TUG or gait speed and disease control after chemotherapy.

The aim of this study was to investigate whether TUG and gait speed measured by the 5-meter walk test (5 mWT) were independent prognostic factors, stronger prognostic factors than PRPF, or predictive factors for disease control in patients with advanced NSCLC receiving carboplatin and vinorelbine in a randomized trial of maintenance pemetrexed therapy.

## 2. Materials and Methods

### 2.1. Patients

From May 2014 to September 2017, 232 patients were enrolled in a randomized controlled trial (RCT) at 19 hospitals in Norway. Eligible patients were treatment naïve, had stage IIIB/IV non-squamous NSCLC (TNM v7), no known activating epidermal growth factor receptor (EGFR) mutation or anaplastic lymphoma kinase (ALK) translocation, World Health Organization Performance Status (WHO PS) 0–2, and adequate bone marrow, liver, and kidney function. Patients who completed four courses of carboplatin/vinorelbine and had WHO PS 0–2 and non-progression were randomized to immediate maintenance pemetrexed therapy or observation followed by pemetrexed at progression. The study closed prematurely due to a stop in patient recruitment when ICI became available in Norway [12].

Patients who received induction chemotherapy and completed TUG and 5 mWT at baseline were analyzed in the present study (Figure 1).

### 2.2. Timed Up and Go Test (TUG)

TUG was performed according to standardized guidelines [9] and registered as the time the patient needed to stand up from a chair, walk 3 m (marked on the floor) at a comfortable pace, turn, walk back, and sit down again. Patients were permitted to use routine walking aids and were instructed not to use their arms to stand up. No physical assistance was given. The task was performed three times, and the average of performances two and three was included in the analyses.

### 2.3. 5-Meter Walk Test (5 mWT)

In the 5 mWT, patients started at zero speed at the starting line, and timing stopped when the patient crossed the line after five meters (marked on the floor). The test was performed at normal speed. Routine walking aids were allowed. The test was performed three times, and the average time of all three performances was included in the analyses [13].

### 2.4. Patient-Reported Physical Function (PRPF)

PRPF was assessed at baseline by the physical functioning scale on the Norwegian version of the European Organization for Research and Treatment of Cancer (EORTC) Quality of Life Questionnaire (QLQ) C30. This score is a compound score of five items: (1) “Do you have any trouble doing strenuous activities, like carrying a heavy shopping bag or a suitcase”, (2) “Do you have any trouble taking a long walk”, (3) “Do you have any trouble taking a short walk outside of the house”, (4) “Do you need to stay in bed or a chair during the day”, and (5) “Do you need help with eating, dressing, washing yourself or using the toilet?” Each item is scored from 0 (not at all) to 4 (very much), summarized, and transformed into a scale ranging from 0 to 100, where a higher score indicates better function [14,15].

### 2.5. Treatment Completion and Endpoints

Treatment completion was assessed in three ways: as the proportion of patients completing all four induction courses, the proportion of patients without any dose reductions of ≥20%, and the proportion without any delays (≥7 days) of induction chemotherapy courses. We also assessed the proportion of patients who were randomized after completion of induction courses, treatment allocation, and number of pemetrexed courses received, as well as the proportion of patients receiving post-study therapy, especially the use of ICIs.

The primary endpoint was overall survival (OS), defined as the time from inclusion (baseline) to death from any cause. The secondary endpoint was disease control, defined as stable disease (SD), partial response (PR), or complete response (CR) according to the RECIST 1.1 [16] evaluated by CT scan 2–3 weeks after the last induction chemotherapy course.

### 2.6. Statistical Considerations

There was no significant difference in overall survival (OS) (*p* = 0.10) between treatment arms in the main trial, and all patients were analyzed as one cohort in the present study [12].

The distribution of TUG and 5 mWT was presented as median and range. There are no established cut off-values for poor TUG or 5 mWT, but 1 m per second or faster is often defined as normal gait speed [17], and 10 s or less has been considered normal values for TUG in previous reports [6,18]. Thus, we considered patients completing the 5 mWT in 5 s or less and those completing the TUG in 10 s or less as having a normal physical function. PRPF was presented as a mean with a 95% confidence interval. The median value was used to separate patients with normal and poor physical function in our analyses. A difference in mean PRPF of 10 was considered clinically significant [19].

Associations between normal or poor physical function (according to TUG, 5 mWT, PRPF), baseline characteristics, and treatment completion were tested with chi-square and Fischer exact test, while the association with age (continuous) was tested with Student’s *t*-test. Scatterplots were used to describe associations between TUG, 5 mWT, and PRPF, and univariable linear regression was used to analyze the strength of any association. The distribution of TUG, 5 mWT, and PRPF according to baseline WHO PS was illustrated with bubble plots. Treatment completion was compared with chi-square and Fischer exact test between patients with normal and poor physical function, while the number of pemetrexed courses was compared with the Mann–Whitney test.

Overall survival (OS) was estimated using the Kaplan–Meier method and compared using the Cox proportional hazard method in uni- and multivariable models. Logistic regression was used for uni- and multivariable analyses of the associations between TUG, 5 mWT, or PRPF and disease control.

All multivariable models were adjusted for baseline characteristics; sex, age (continuous), stage of disease (III versus IV), and WHO PS (0, 1, and 2). TUG, 5 mWT, and PRPF were entered separately in multivariable analyses both as continuous and dichotomous variables. In exploratory analyses, the multivariable model of PRPF and OS was adjusted for TUG and 5 mWT, respectively, and another model of PRPF and OS was adjusted for receipt of ICIs.

A two-sided *p* < 0.05 was considered statistically significant. SPSS v27 was used for all statistical analyses. Plots were made in SPSS or RStudio v1.4.

### 2.7. Approvals

The RCT was approved by the Regional Committee for Medical Research Ethics in Central Norway and The Norwegian Medicines Agency. ClinicalTrials.gov identifier: NCT02004184.

## 3. Results

### 3.1. Baseline Characteristics

Of the 232 patients included in the RCT, 208 (90%) performed TUG and 5 mWT at baseline and were included in the present study. Among these, the median age was 67 years (range 46–83), 112 (54%) were women, 195 (94%) had stage IV disease, and 66 (32%), 112 (54%), and 30 (14%) had WHO PS 0, 1, and 2, respectively. There were more patients with WHO PS 2 among those with poor physical function according to TUG (*p* < 0.01), 5 mWT (*p* < 0.01), and PRPF (*p* < 0.01) (Table 1).

### 3.2. Treatment Completion

Of 208 patients, 146 (70%) received all four induction courses. Patients with a TUG ≥10 s were less likely to complete all four courses (≥10 s: 58%, <10 s: 74%; *p* < 0.01), but there were no associations between 5 mWT (*p* = 0.34) or PRPF (*p* = 0.08) and completion of four courses (Table 1).

In total, 95 (46%) patients had at least one dose reduction, and 72 (35%) patients had at least one chemotherapy course delayed. There were no differences in dose reductions (TUG: *p* = 0.77, 5 mWT: *p* = 0.60, and PRPF: *p* = 0.68) or delays between patients with normal or poor physical function (TUG: *p* = 0.17, 5 mWT: *p* = 0.64, and PRPF: *p* = 0.77).

Of all patients, 55 (26%) received all four induction courses without any dose reductions or delays, and there were no differences between patients with normal or poor physical function (TUG: *p* = 0.11, 5 mWT: *p* = 0.44, PRPF: *p* = 0.49).

Only 97 (47%) were randomized after completion of induction chemotherapy, 50 (24%) to immediate maintenance pemetrexed therapy (median 3 courses, range 0–29), and 47 (23%) to the control arm, of whom 34 (72%) patients received pemetrexed at progression (median 4 courses, range 1–12). Patients with a normal physical function according to TUG were more likely to be randomized (*p* < 0.01), while there were no significant associations with 5 mWT (*p* = 0.26) or PRPF (*p* = 0.06) (Table 1). Allocation to treatment arm was balanced (TUG: *p* = 0.39, 5 mWT: *p* = 0.52, PRPF: *p* = 1.00) and there were no differences in number of pemetrexed courses received between patients with normal or poor physical function (TUG: *p* = 0.90, 5 mWT: *p* = 0.93, PRPF, *p* = 0.62).

In total, 114 (55%) of the patients received post-study treatment, of whom 45 (22%) received ICI therapy. There was no difference in proportions of patients receiving salvage therapy between those with normal or poor physical function (TUG: *p* = 0.37, 5 mWT: *p* = 0.98, and PRPF: *p* = 0.20). However, patients with a normal PRPF were more likely to receive an ICI (≥73.3: 28%, <73.3: 17%, *p* = 0.01), while no such associations were observed for TUG (*p* = 0.09) or 5 mWT (*p* = 0.35) (Table 1).

### 3.3. Timed Up and Go (TUG)

The median TUG was 7.8 s (range 0.7–44.2 s). Forty-two (20%) patients had TUG ≥ 10 s. There was no difference between men and women or patients with stage IIIB or IV disease. Patients with a poor WHO PS had a longer TUG: WHO PS 0: median 7.2 s (range 2.6–19.6 s), WHO PS 1: median 7.8 s (range 0.7–13.5 s), and WHO PS 2: median 10.7 s (range 6.8–44.2 s) (Table 1). The association between WHO PS and TUG is illustrated in Figure 2. The largest variation in TUG was observed among patients with WHO PS 2.

### 3.4. 5-Meter Walk Test (5 mWT)

The median 5 mWT was 4.5 s (range 1.8–28.1 s). Seventy-seven (37%) had 5 mWT ≥ 5 s. There was no difference between men and women or patients with stage IIIB or IV disease. Patients with poor WHO PS had a longer 5 mWT: WHO PS 0: median 4.4 s (range 1.8–14.0 s), WHO PS 1: median 4.1 s (range 2.1–14.0 s), and WHO PS 2: median 6.1 s (range 3.0–28.1 s) (Table 1). The associations between WHO PS and 5 mWT are illustrated in Figure 2B. The largest variation in 5 mWT was observed among patients with WHO PS 2.

### 3.5. Patient-Reported Physical Function (PRPF)

The QLQ C30 was completed at baseline by 173 (83%) patients. The mean PRPF was 72.2 (95% CI 69.3–75.2), and the median was 73.3. There was no significant difference in mean PRPF between men and women, but patients with stage IIIB reported better PRPF than patients with stage IV (83.0 vs. 71.5). Patients with a poor WHO PS had a lower mean PRPF: WHO PS 0: 78.5, WHO PS 1: 72.7, and WHO PS 2: 58.8 (Table 1). The associations between WHO PS and PRPF are illustrated in Figure 2C. There was a large variation in PRPF among patients independent of WHO PS.

### 3.6. Association between TUG, 5 mWT, PRPF, and WHO PS

A worse TUG and 5 mWT was significantly associated with lower PRPF, but variation in physical tests only partly explained the variation in PRPF (TUG versus PRPF: R2 = 0.11, *p* < 0.01; 5 mWT versus PRPF: R2 = 0.10, *p* < 0.01). The association between TUG and 5 mWT was stronger (R2 = 0.23, *p* < 0.01) (Figure 3D–F). Several patients with good physical function according to TUG or 5 mWT reported a low PRPF (Figure 3D,E).

There was less variation in TUG or 5 mWT among patients with WHO PS 0 and 1 than among patients with WHO PS 2, but a wide range of values was observed for all WHO PS categories (Figure 2A,B).

### 3.7. Overall Survival

Median OS was 10.0 months (95% CI 8.82–11.18) for the whole study population. Median follow up time was 36.2 months (95% CI 33.0–39.3). Neither sex (*p* = 0.42) nor stage of disease (*p* = 0.14) was significantly associated with survival. A poor WHO PS was significantly associated with shorter survival both in uni- and multivariable models (PS 2 vs. 0: *p* < 0.01) (Table 2). When entered as continuous variables in univariable models, poor physical function was associated with shorter survival: TUG: HR 1.05, *p* < 0.01, 5 mWT: HR 1.05, *p* = 0.03, PRPF: HR 1.01, *p* < 0.01. In multivariable analysis, only PRPF remained an independent prognostic factor: TUG: *p* = 0.18, 5 mWT: *p* = 0.13, PRPF: HR 1.01, *p* = 0.03 (Table 2). In exploratory analyses, the association between PRPF and survival reached borderline significance when the multivariable model was adjusted for TUG (HR 1.01, *p* = 0.05) or 5 mWT (HR 1.01, *p* = 0.05), but not when it was adjusted for post-study ICI therapy (HR 1.00, *p* = 0.42).

When patients were categorized as having normal or poor physical function, TUG (*p* < 0.01) but not 5 mWT (*p* = 0.21) was significantly associated with survival. In multivariable analyses, neither TUG (*p* = 0.07) nor 5 mWT (*p* = 0.41) were significantly associated with survival. In contrast, a normal PRPF was significantly associated with improved survival both in uni- (HR 1.80, *p* < 0.01) and multivariable analyses (HR 1.60, *p* < 0.01) (Table 3 and Figure 4).

### 3.8. Disease Control

Treatment response was evaluated in 179 (86%) patients. Overall, disease control was achieved in 109/179 (61%) patients and in 92/149 (62%) patients that completed PRPF at baseline. TUG, 5 mWT, or PRPF were not statistically significant predictors for disease control in uni- (TUG: *p* = 0.66, 5 mWT: *p* = 0.36, and PRPF: *p* = 0.94) or multivariable analyses (TUG: *p* = 0.30, 5 mWT: *p* = 0.69, and PRPF: *p* = 0.13) (Appendix A).

## 4. Discussion

In this study of patients included in our trial of maintenance pemetrexed therapy in advanced non-squamous NSCLC, physical function as measured by TUG and 5 mWT were not independent prognostic factors for survival, while patient-reported physical function (PRPF) was. Patients with a good physical function measured by TUG were more likely to complete four courses and thus be randomized, but none of the measures of physical function were significantly associated with achieving disease control at evaluation after induction chemotherapy.

To the best of our knowledge, this is the first study investigating whether the measured physical function is an independent prognostic factor in patients with advanced non-squamous NSCLC receiving palliative platinum-doublet chemotherapy, including analyses adjusting for established prognostic factors, baseline characteristics, and treatment completion.

Several studies have investigated the associations between TUG or gait speed and OS in patients with cancer, including two recent systematic reviews (Verweji et al. in 2016 and Ezzatvar in 2020); the latter also included a meta-analysis [7,8]. Seven of the studies included in these reviews analyzed patients with NSCLC, with a proportion ranging between 8 and 100% [6,11,20,21,22,23,24].

Four of these studies analyzed TUG in patients with advanced NSCLC that received chemotherapy [6,11,20,21]. As in our study, TUG was a prognostic factor for survival in univariable analyses. However, in the only study including multivariable analyses, TUG was also found to be an independent prognostic factor. In contrast to our cohort, only 28 out of 348 patients had NSCLC in that study [11].

In studies investigating gait speed, the association with survival is less consistent [22,23,24,25], and gait speed was not an independent prognostic factor in a study (n = 112) in which 24% of patients had lung cancer, 44% stage III-IV disease, and 26% received palliative chemotherapy [24]. However, differences in patient selection and the use of different tests for measuring gait speed make it difficult to compare results across studies. To the best of our knowledge, no previous studies have investigated the associations between TUG, 5 mWT, or PRPF and disease control.

The fact that objectively assessed physical function was not an independent prognostic factor in our study cohort might be explained by a selection bias; all patients were considered fit for palliative chemotherapy in the setting of an RCT. The median age (67 years) and rate of PS 2 (14%) were lower than seen in the daily clinic. Overall, there was little variation in TUG and 5 mWT, and median TUG (7.8 s) was lower than reported in community-dwelling adults of similar age, height, and weight (9.0 s) [26]. Patients in our study reported relatively high physical function compared to other populations with advanced NSCLC [27]. The limited variation and overall good physical function might have limited the chances of detecting clinically relevant associations.

The fact that PRPF was an independent prognostic factor is consistent with previous studies [5,6], and it might be that PRPF better reflects changes in physical function or how the physical function is compared with the patients’ former or habitual daily function. Consequently, it may be more sensitive to disease-specific changes and, thus, holds more prognostic information than TUG and 5 mWT. Interestingly, many patients with good physical function according to TUG or 5 mWT reported a poor PRPF, indicating that PRPF includes other aspects than TUG and 5 mWT. Patients with a poor WHO PS were more likely to report a poor PRPF, but the prognostic information from PRPF was independent of WHO PS in multivariable analyses.

Despite being the largest of its kind, this study is still limited by size. Although patients received the same first-line treatment, differences in post-study therapy might have influenced our results. ICI therapy for advanced NSCLC was introduced in Norway during the enrolment period, and the availability of ICI varied with time and between hospitals. Patients with a normal PRPF were more likely to receive ICI therapy, and fitness for such treatment might explain the improved survival among these patients, supported by the fact that PRPF was no longer an independent prognostic factor when adjusting for the use of ICI in the exploratory multivariable model.

Another possible limitation is that not all patients completed the physical functions tests or reported their physical function. However, we believe that a completion rate for physical tests of 90% is quite good in a multicenter RCT, and patients with missing data did not differ from other patients with respect to age, gender, WHO PS, or disease stage (data not shown). Our results are based on an RCT from the pre-ICI era, but they are still relevant since many patients with advanced NSCLC still receive platinum-doublet chemotherapy, either combined with ICIs in the first-line setting or as salvage therapy.

In conclusion, measuring TUG and 5 mWT did not provide clinically relevant predictive or prognostic information in patients with advanced non-squamous NSCLC receiving platinum-doublet chemotherapy. TUG and 5 mWT held less prognostic information than physical function (PRPF) reported by patients on the EORTC QLQ C30.

## Figures and Tables

**Figure 1 healthcare-10-00922-f001:**
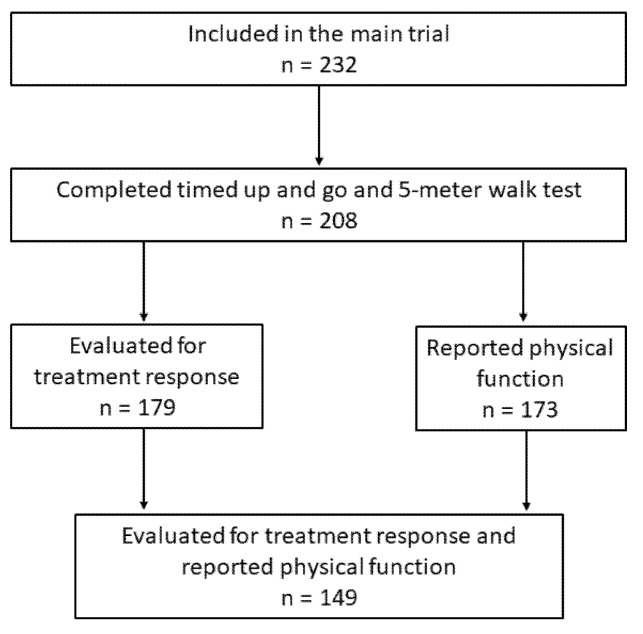
Patient selection.

**Figure 2 healthcare-10-00922-f002:**
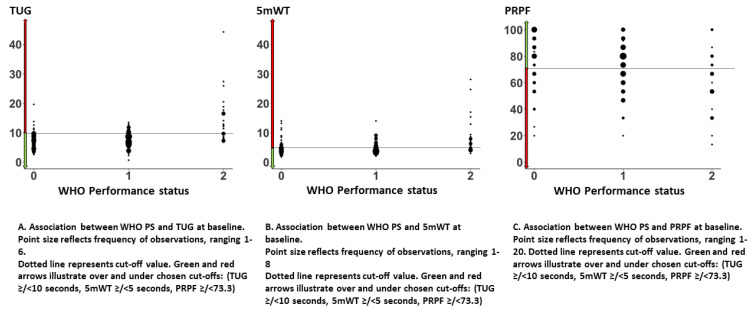
Associations between WHO performance status and timed up and go, 5-meter walk test, and patient-reported physical function.

**Figure 3 healthcare-10-00922-f003:**
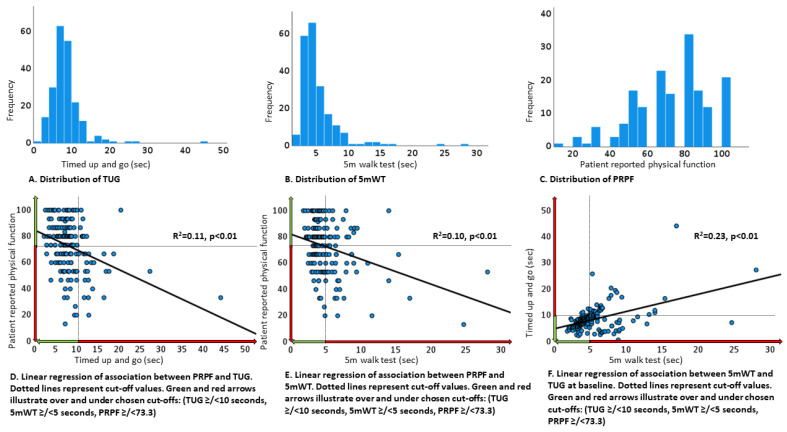
Physical performance tests and patient-reported physical function at baseline.

**Figure 4 healthcare-10-00922-f004:**
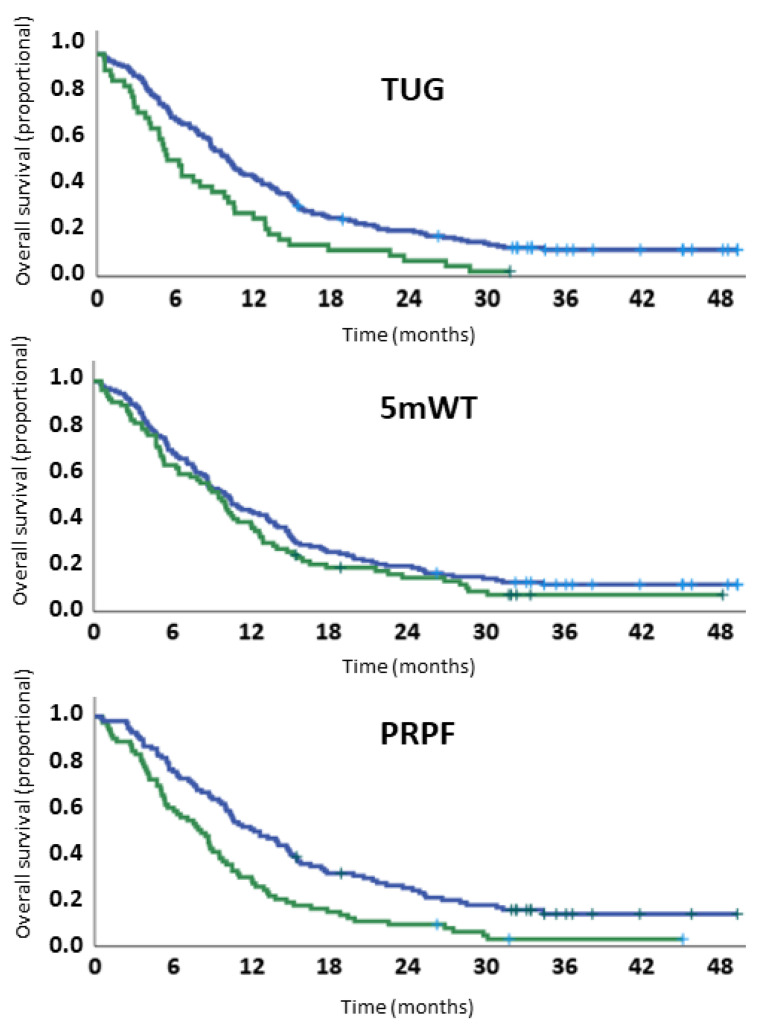
Survival curves according to cutoff values for measured and patient-reported physical function.

**Table 1 healthcare-10-00922-t001:** Patient and treatment characteristics.

				TUG						5 mWT						PRPF					
					<10 s		≥10 s				<5 s		≥5 s				≥73.3		<73.3		
		*n*	%	Median (Range)	*n*	%	*n*	%	*p*	Median (Range)	*n*	%	*n*	%	*p*	Mean (95% CI)	*n*	%	*n*	%	*p*
**TUG**		208	(100%)	7.8 (0.7–44.2)	166	(80%)	42	(20%)													
**5 mWT**		208	(100%)							4.5 (1.8–28.1)	131	(63%)	77	(37%)							
**PRPF**		173	(83%)													72.2 (69.3–75.2)	100	(58%)	73	(42%)	
**Age**	**Median (range)**	67	(46–83)		66	(46–83)	69	(55–83)	0.03		66	(46–83)	69	(51–83)	0.03		68	(66–69)	66	(65–67)	0.17
**Sex**	**Male**	96	(46%)	8.0 (2.6–27.4)	78	(47%)	18	(43%)		4.2 (1.8–28.1)	62	(47%)	34	(44%)		75.9 (71.8–79.9)	48	(48%)	31	(42%)	
	**Female**	112	(54%)	7.6 (0.7–44.2)	88	(53%)	24	(57%)	0.63	4.5 (2.0–24.7)	69	(53%)	43	(56%)	0.66	69.2 (65.0–73.3)	52	(52%)	42	(58%)	0.47
**Stage**	**IIIB**	13	(6%)	7.7 (3.6–10.0)	13	(15%)	-	-		4.2 (2.6–9.5)	9	(7%)	4	(5%)		83.0 (74.2–91.8)	9	(9%)	2	(3%)	
	**IV**	195	(94%)	7.8 (0.7–44.2)	153	(85%)	42	(100%)	0.08	4.5 (1.8–28.1)	122	(93%)	73	(95%)	0.63	71.5 (68.4–74.6)	91	(91%)	71	(97%)	0.10
**WHO PS**	**0**	66	(32%)	7.2 (2.6–19.6)	58	(35%)	8	(19%)		4.4 (1.8–14.0)	44	(34%)	22	(29%)		78.5(72.9–84.2)	36	(36%)	16	(22%)	
	**1**	112	(54%)	7.8 (0.7–13.5)	93	(56%)	19	(45%)		4.1 (2.1–14.0)	77	(59%)	35	(45%)		72.7 (69.3–76.2)	56	(56%)	37	(51%)	
	**2**	30	(14%)	10.7 (6.8–44.2)	15	(9%)	15	(36%)	<0.01	6.1 (3.0–28.1)	10	(7%)	20	(26%)	<0.01	58.8 (50.3–67.3)	8	(8%)	20	(27%)	<0.01
**Completed** 4 **induction courses**	**No**	62	(30%)	8.2 (0.7–25.9)	43	(26%)	19	(42%)		4.6 (2.5–24.7)	36	(27%)	26	(34%)		67.8 (61.6–73.9)	26	(26%)	28	(38%)	
	**Yes**	146	(70%)	7.7 (2.3–44.2)	123	(74%)	23	(58%)	0.01	4.3 (1.8–28.1)	95	(63%)	51	(66%)	0.34	74.2 (71.0–77.5)	74	(74%)	45	(62%)	0.08
**Randomization**	**No**	111	(53%)	8.2 (0.7–44.2)	80	(48%)	31	(74%)		4.6 (1.8–24.7)	66	(50%)	45	(58%)		69.3 (65.1–73.6)	46	(46%)	44	(60%)	
	**Yes**	97	(47%)	7.7 (2.3–27.4)	86	(52%)	11	(27%)	<0.01	4.2 (2.0–28.1)	65	(50%)	32	(42%)	0.26	75.4 (71.3–79.4)	54	(54%)	29	(40%)	0.06
	**-Observation**	47	(23%)	7.7 (2.9–12.6)	43	(26%)	4	(10%)		4.2 (2.5–11.6)	30	(23%)	17	(22%)		75.2 (68.6–81.9)	26	(26%)	14	(19%)	
	**-Maintenance**	50	(24%)	7.4 (2.3–27.4)	43	(26%)	7	(17%)	0.39	4.3 (2.0–28.1)	35	(27%)	15	(20%)	0.52	75.5 (70–5–80.5)	28	(28%)	15	(21%)	1.00
**Post-study** **immunotherapy**	**No**	163	(78%)	7.9 (0.7–44.2)	126	(76%)	37	(88%)		4.5 (1.8–28.1)	100	(76%)	63	(82%)		69.9 (66.6–73.2)	72	(72%)	64	(83%)	
	**Yes**	45	(22%)	7.0 (2.3–12.6)	40	(24%)	5	(12%)	0.09	4.2 (2.1–13.2)	31	(24%)	14	(18%)	0.35	80.9 (75.0–86.7)	28	(28%)	9	(17%)	0.01

**Table 2 healthcare-10-00922-t002:** Survival analyses.

				Univariable Analysis			Multivariable Model with TUG			Multivariable Model with 5 mWT			Multivariable Model with PRPF		
		*n*	(%)	HR	95% CI	*p*-Value	HR	95% CI	*p*-Value	HR	95% CI	*p*-Value	HR	95% CI	*p*-Value
**TUG ***		208	(100%)	1.05	1.02–1.08	<0.01	1.03	0.99–1.07	0.18						
**5 mWT ***		208	(100%)	1.05	1.01–1.10	0.03				1.04	0.99–1.09	0.13			
**PRPF ***		173	(83%)	1.01	1.01–1.02	<0.01							1.01	1.00–1.02	0.03
**Age ***		208	(100%)	1.00	0.98–1.02	0.94	0.99	0.97–1.01	0.99	0.99	0.97–1.01	0.99	0.96	0.97–1.02	0.67
**Sex**	**Male**	96	(46%)	1			1			1			1		
	**Female**	112	(54%)	1.13	0.84–1.50	0.42	1.09	0.81–1.45	0.58	1.08	0.81–1.45	0.61	1.06	0.76–1.49	0.73
**Stage of disease**	**IIIB**	13	(6%)	1			1			1			1		
	**IV**	195	(94%)	1.55	0.86–2.78	0.14	1.51	0.82–2.76	0.19	1.50	0.82–2.74	0.19	1.21	0.65–2.28	0.55
**WHO PS**	**0**	66	(32%)	1			1			1			1		
	**1**	112	(54%)	1.45	1.04–2.02	0.03	1.51	1.07–2.12	0.02	1.56	1.11–2.19	0.01	1.38	0.94–2.02	0.10
	**2**	30	(14%)	2.57	1.63–4.06	<0.01	2.25	1.32–3.83	<0.01	2.44	1.51–3.96	<0.01	2.11	1.23–3.62	<0.01

* Entered as a continuous variable. TUG—timed up and go; 5 mWT—5-meter walk test; PRPF—patient-reported physical function.

**Table 3 healthcare-10-00922-t003:** Differences in survival according to cutoff values for measured and patient-reported physical function.

	Median OS	95% CI	HR Univariable Model	95% CI	*p*	HR Multivariable Model	95% CI	*p*
TUG < 10 sek	10.4	8.6–12.2	1			1		
TUG ≥ 10 sek	6.3	3.9–8.7	1.74	1.23–2.47	<0.01	1.43	0.97–2.10	0.07
5 mWT < 5 sek	10.4	8.4–12.3	1			1		
5 mWT ≥ 5 sek	9.6	7.6–11.5	1.21	0.90–1.63	0.21	1.14	0.84–1.55	0.41
PRPF ≥ median	12.2	8.6–15.6	1			1		
PRPF < median	8.2	6.1–10.2	1.8	1.31–2.49	<0.01	1.6	1.14–2.24	<0.01

## Data Availability

The data that support the finding of this study are available from the corresponding author; KS, upon reasonable request.

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
