# Peer review of "Associations between Measured and Patient-Reported Physical Function and Survival in Advanced NSCLC"

_healthcare, 2022, doi:10.3390/healthcare10050922_

Round 1
Reviewer 1 Report
The paper is very well written and heavily data-driven. I have some minor comments -
- Intro line 52 : what % of these studies have been used in lung cancer?
- Flow chart on page 79 :
These numbers dont add up to 208 - the authors should signify of any overlaps between each of the groups.
3. For results 3.3 to 3.5 it would be nice to have some concluding sentences.
Author Response
The paper is very well written and heavily data-driven. I have some minor comments:
- Intro line 52: what % of these studies have been used in lung cancer?
Response: There was a considerable variation between studies in the proportion of lung cancer patients included (8-100%), and this is addressed in detail in the discussion. We have made minor adjustments to the text in response to the question (Lines 359-360 and 366-367). - Flow chart on page 79. These numbers don’t add up to 208 - the authors should signify of any overlaps between each of the groups.
Response: We have improved the flow chart (Figure 1) to better clarify the different cohorts in the study.
- For results 3.3 to 3.5 it would be nice to have some concluding sentences.
Response: We included concluding sentences in paragraph 3.3, 3.4, and 3.5.: Lines 212-213, 223-224 and 231-232.
Reviewer 2 Report
In this article, Stokke and coauthors investigate whether measured physical performance was an independent or stronger prognostic factor than Patient-reported physical function (PRPF) in patients with advanced (stage III/IV) NSCLC receiving platinum doublet chemotherapy. They demonstrated that Timed up and go (TUG) and gait speed measured by the 5-meter walk test (5mWT) were independent and stronger prognostic factors than PRPF, or predictive factors for disease control in this cohort of NSCLC patients.
The manuscript is well written. It flows logically and the data are properly presented. References are updated and adequate.
Major comments:
The patients in the cohort have non-squamous NSCLC. Did the Authors find any differences in their analysis comparing patients with adenocarcinoma and large cell carcinoma?
Did the Authors find any differences in their analysis comparing smoking and not smoking patients?
Were the patients enrolled in this study at the same stage of the treatment when they performed the TUG and 5mWT? At which stage they were?
In Table 1:
Please, clarify the “randomization”. What is the difference between “randomization” “no” 111 and 163 patients.
Please, clarify which immunotherapic treatment the patients (45 patients) received.
Minor comments:
The legends below the graphs in figure 2 and 3 are small to be read
Author Response
The manuscript is well written. It flows logically and the data are properly presented. References are updated and adequate
- The patients in the cohort have non-squamous NSCLC. Did the Authors find any differences in their analysis comparing patients with adenocarcinoma and large cell carcinoma?
Response: Most patients in this study had adenocarcinoma, with only one patient with large cell carcinoma. Consequently, we did not explore differences between patients with different histology. - Did the Authors find any differences in their analysis comparing smoking and not smoking patients?
Response: Due to a limited sample size, our analyses were adjusted for a limited set of established prognostic and predictive factors, including (sex, age, stage of disease, WHO PS). In preparing our response to the reviewers we performed explorative analysis of smoking status. There were no differences in TUG, 5mWT or PRPF between smokers and non-smokers, and smoking status was not a prognostic factor in multivariable analysis of OS or a predictive factor in multivariable analysis of disease control. - Were the patients enrolled in this study at the same stage of the treatment when they performed the TUG and 5mWT? At which stage they were?
Response: All patients were treatment naïve and performed tests for TUG and 5mWT prior to starting the same first-line chemotherapy in the setting of a RCT. This is explained in the method section part 2.1; line 68, and lines 77-78. - In Table 1: Please, clarify the “randomization”. What is the difference between “randomization” “no” 111 and 163 patients.
Response: Due to a formatting error, “no 163” (belonging to the subheading for post-study treatment) was placed under “randomization”. The table is now corrected. - Please, clarify which immunotherapic treatment the patients (45 patients) received.
Response: As stated in the manuscript (lines 200-201), 45 patients received immune checkpoint inhibitors (ICI) in later lines of treatment, but unfortunately this study was not designed to collect more detailed information on the drug of choice. The majority is however expected to have received the PD1 inhibitor pembrolizumab as this became available according to our national guidelines. We are not able to give more detailed information on the use of immunotherapy. However, we do not believe this information to be essential to the interpretation of our results. - The legends below the graphs in figure 2 and 3 are small to be read
Response: We have increased the font size to improve readability.
Reviewer 3 Report
Nicely written manuscript. Minor edits were suggested, see attached.

Author Response
Nicely written manuscript. Minor edits were suggested.
Response: We have changed the text according to your edits. Please see corrections marked by “track changes” in the article. (line(s) 60, 75-76, 98-102, 237, 405-408).